# Exploring Deep Recurrent Models with Reinforcement Learning for Molecule Design

## Abstract

The design of small molecules with bespoke properties is of central importance to drug discovery. However significant challenges yet remain for computational methods, despite recent advances such as deep recurrent networks and reinforcement learning strategies for sequence generation, and it can be difficult to compare results across different works. This work proposes 19 benchmarks selected by subject experts, expands smaller datasets previously used to approximately 1.1 million training molecules, and explores how to apply new reinforcement learning techniques effectively for molecular design. The benchmarks here, built as OpenAI Gym environments, will be open-sourced to encourage innovation in molecular design algorithms and to enable usage by those without a background in chemistry. Finally, this work explores recent development in reinforcement-learning methods with excellent sample complexity (the A2C and PPO algorithms) and investigates their behavior in molecular generation, demonstrating significant performance gains compared to standard reinforcement learning techniques.

## 1 Introduction

Novel drugs are developed using *design - make - test* cycles: molecules are designed, synthesized in the laboratory, and then tested for their biological effect. The insights gained from these tests then inform the design for the next iteration. The objective of *de novo design* methodologies is to perform this cycle with computational methods (Brown, 2015; Schneider, 2013). The *test* phase was the first to be automated, using the broad categorization of machine learning models known as quantitative structure-activity/property relationships (QSAR/QSPR) to predict the activity of a molecule against a certain biological target, or physicochemical properties. To *make* virtual molecules, symbolic approaches based on graph rewriting have been used, which are domain-specific and rely on extensive hand-engineering by experts. To optimize the properties of a molecule, for example its activity against a biological target (*design*), global optimization approaches such as evolutionary algorithms or ant colony optimization have been used (Brown, 2015; Schneider, 2013). Symbolic approaches have been highlighted as either generating unrealistic molecules that would be difficult to synthesize, or for being too conservative, and therefore not sufficiently exploring the space of tractable molecules (Schneider, 2013; Brown & Boström, 2016).

Recently, generative models have been proposed to learn the distribution of real druglike molecules from data, and then to generate chemical structures that are appropriate for the application domain (White & Wilson, 2010). Interestingly, the generation of molecules is related to natural language generation (NLG). Two classic problems of NLG – preserving coherent long-range dependencies, and syntactic and semantic correctness – directly map to molecules. Current investigations draw heavily from tools developed for language tasks, including variational autoencoders (VAE) (Gómez-Bombarelli et al., 2016; Kusner et al., 2017), recurrent neural network (RNN) models (Segler et al., 2017; Jaques et al., 2017; Olivecrona et al., 2017), generative adversarial networks (GAN) (Guimaraes et al., 2017) and Monte Carlo Tree Search (MCTS) (Yang et al., 2017).

This work seeks to consolidate the growing body of recurrent models for molecular design that employ reinforcement learning. Here, we suggest a set of 19 benchmarks of relevance to *de novo* design. Furthermore, an implementation of these benchmarks as an OpenAI Gym is provided to the community to spur further innovation. Finally, we demonstrate state-of-the-art performance using new techniques drawing from recent advances in reinforcement learning.

CC(=O)Nc1ccc(O)cc1

Paracetamol

Figure 1: Molecular graphs can be represented as strings using the SMILES notation. Letters correspond to element symbols, rings opening and closing is indicated with numbers, and branching with round brackets.

## 2 REPRESENTING MOLECULES AS SEQUENCES

*De novo* design can be seen as a structured prediction problem, where molecular structures have to be predicted. This paper will use the term "molecule" to denote the chemical structures of interest. Molecular structures are represented well as labeled graphs $M = (A, B)$, with atoms $A$ as vertices and bonds $B$ as edges, elemental types as vertex labels, and bond order as edge labels (Brown, 2009). While neural network models which output graphs remain underexplored, sequence generation is well established. To encode molecules as sequences of symbols (strings), the canonicalized Simplified Molecular-Input Line-Entry System (SMILES) notation (Weininger, 1988; Weininger et al., 1989) can be used (see Fig. 1). This establishes the link to sequence-based language-focused neural network models (Segler et al., 2017; Goldberg, 2016), which can be then be used to generate molecules.

## 3 DATA AND BENCHMARKS

### 3.1 DATA

In this paper, we expand the scope of previous work to a standardized, much larger dataset, which furthermore is of real-world interest to drug discovery scientists. Here, we build on the ChEMBL-23 dataset (Gaulton et al., 2011), a collection extracted from the scientific literature of 1,735,442 distinct compounds and their reported biological activities on 11,538 targets. Though partitioned and filtered, this work employs substantially more training examples than previous work to access the breadth of chemistries that have been demonstrated to be of interest. The preprocessing steps can be found in the Appendix.

To stimulate further work in this domain, our OpenAI Gym interface to these benchmarks will be open-sourced to allow the community to prototype new RL algorithms for chemistry.[1] Here, we offer 19 benchmarks to be used for molecule generation, comprised of basic suitability benchmarks, basic physicochemical property optimizations, drug-likeness approximations, and multi-objective balancing. The benchmark framework is general enough to be used with any possible small molecule generation method, whether rule-based or learned, and is not limited to sequence-based generation relying on SMILES.

### 3.2 VALIDITY AND DIVERSITY OF GENERATED MOLECULES

A basic but crucial molecular generation benchmark is simply what percentage of the sampled molecules are valid. For a sample set $S$ of cardinality $m$ sampled from a model $M$, the percentage of valid molecules is:

$$Y_i \sim \pi_\theta; \quad S = \{Y_1, \ldots, Y_m\}; \quad R_{valid} = \frac{1}{m} \sum_i^m \text{valid}(Y_i); \quad R_{valid} \in [0, 1] \qquad (1)$$

where valid() is a function returning 1 if the open-source chemoinformatics toolkit RDKit (Landrum et al.) is able to return a valid molecular object given the SMILES representation, and 0 otherwise.

Similarly, it is not ideal if the generative model is able to produce a valid molecule, but only repeats the same molecule. Therefore, the *ratio unique* benchmark samples $m$ molecules from the model,

---

[1] A link will be provided after review to preserve anonymity.

Figure 2: Four of ten target molecules for the Tanimoto benchmark. See Supplementary Material for the full set.

and measures the number of unique molecules:

$$Y_i \sim \pi_\theta; \quad R_{unique} = \frac{1}{m} \left| \bigcup_{i=1}^{m} \{Y_i\} \right|; \quad R_{unique} \in [0,1] \tag{2}$$

### 3.3 SINGLE OBJECTIVE MAXIMIZATION

Previous work has explored a variety of optimization objectives of interest to molecular designers. Physicochemical properties, such as the octanol-water partition coefficient (ClogP) and molecular weight (MW), have strong implications for the viability of a molecule progressing as a potential drug and have been explored as optimization objectives in previous work (Firth et al., 2015; Jaques et al., 2017; Gómez-Bombarelli et al., 2016). Similarly, SMARTS (SMILES Arbitrary Target Specification) sequences allow to specify substructures (subgraphs) which should be contained in the target molecules. This allows to define a reward function SMARTS(X, Y) which returns 1 if a generated molecule $Y$ contains subgraph $X$ and $-1$ otherwise. The RDKit (Landrum et al.) includes automatic code to calculate these functions, providing a straightforward route to include these objectives in RL scenarios. For a target value $x$ or respectively subgraph $X$, and a generated sequence $Y$, the reward is:

$$R_{LogP}(x,Y) = \frac{1}{25}(x - \text{LogP}(Y))^2 + 1; \quad R_{LogP}(x,Y) \in [-\infty, 1] \tag{3}$$

$$R_{MW}(x,Y) = \frac{1}{10^5}(x - \text{MW}(Y))^2 + 1; \quad R_{MW}(x,Y) \in [-\infty, 1] \tag{4}$$

$$R_{SMARTS}(X,Y) = \text{SMARTS}(X,Y); \quad R_{SMARTS}(x,Y) \in \{-1, 1\} \tag{5}$$

This work chooses five ClogP points $x \in \{-1, 0, 1, 2, 3\}$ as benchmarks.

A more challenging optimization objective is to generate a family of molecules similar to a target given only a fingerprint of the target molecule, in an unpopulated area of chemical space. Here, the commonly-used Functional Connectivity Fingerprint Counts (FCFC4; Rogers & Hahn (2010)) is used to encode the molecular graph as a fixed-size integer vector. After encoding both the proposed molecule and the target molecule, their Tanimoto (akin to Jaccard) similarity is calculated. This work uses ten marketed drugs which modulate different biological target types (Figure 2, Table 3) as molecular targets for approximation. To ensure that no leakage occurs between test and train set, the target molecules themselves as well as compounds similar to them were removed from the training data (see Appendix). The reward of the Tanimoto similarity with $x, y$ as the fingerprint vectors of $X, Y$ is:

$$R_{Tani}(X,Y) = 2\frac{\sum_{i=1}^{n} x_i y_i}{\sum_{i=1}^{n} (x_i^2 + y_i^2 - x_i y_i)} - 1; \quad R_{Tani}(X,Y) \in [-1, 1] \tag{6}$$

### 3.4 MULTI-OBJECTIVE MAXIMIZATION

To test balanced optimization, two further tests are proposed: Lipinski's Rule-of-Five (Ro5) parameters for the estimation of solubility and permeability important for oral bioavailability, and a weighted multi-parameter optimization. The Ro5 is a heuristic for druglikeness, evaluating four parameters that can be calculated readily from a molecular structure: molecular weight (MW), ClogP, hydrogen-bond donors (HBD) and acceptors (HBA) (Ghose et al., 1999; Lipinski, 2004) In this work, the Ro5 formulation used is to penalize (a) *via* MSE, the MW if outside the interval [180, 500]; (b) *via* MSE, the ClogP if outside [-0.4, 5.6]; (c) by absolute error, the number of HBDs if greater than 5;

(d) by absolute error, the number of HBAs if greater than 10. The final penalties are summed, scaled by 1e-3, and added to 1 to place the reward $R_{Ro5} \in [-\infty, 1]$.

Similarly, any arbitrary balanced weighting of objective functions can be used to simulate a multi-objective optimization, similar to those that are recognized in drug discovery scenarios. Here, an equally-weighted target of ClogP=4, SMARTS fragment-matching to include a benzene ring, and a MW of 180 is defined.

$$R_{MPO} = \frac{1}{3}(R_{LogP}(4, Y) + R_{SMARTS}('\texttt{c1ccccc1}', Y) + R_{MolWt}(180, Y)) \tag{7}$$

$$R_{MPO} \in [-\infty, 1] \tag{8}$$

## 4 MODELS

### 4.1 RECURRENT NEURAL NETWORKS

Gated recurrent models such as Long short-term memory (LSTM) (Hochreiter & Schmidhuber, 1997) are important for modern deep RNN models, and are used as the principal deep model used in this work (Graves, 2013). These RNN models are able to process the SMILES format strings with the addition of an embedding layer as is commonly used in natural language tasks (Gal & Ghahramani, 2016; Cho et al., 2014). Fully-connecting the output of the RNN model to a layer of neurons equal in cardinality to the vocabulary size of our SMILES language (42 symbols, see Supplementary Material Table 2) can be trained to produce a probability distribution over the output symbols. A model $\pi_\theta$ parameterized by $\theta$ trained on sequences $z$ of length $T$ in dataset $S$ can be trained with the following standard differentiable loss (the cross-entropy loss):

$$L^{MLE}(\theta) = -\sum_{t=1}^{T} \log \pi_\theta(z_t | z_{1:t-1}); \quad z = (z_1, \ldots, z_T) \in S \tag{9}$$

In this work, this is called the *pretrained maximum likelihood estimation (MLE) model* and can be trained using standard supervised gradient descent techniques with backpropagation (Goldberg, 2016). This model functions as a composable, initialized block for more advanced architectures due to its effectiveness in generating realistic chemical sequences. Algorithm 1 demonstrates a slightly more unusual formulation in which a pretrained MLE model $\pi_\theta$ is combined with an arbitrary reward function $R$ to maximize reward by alternately sampling and retraining via Eq. 9 on the $k$-highest reward sequences (Segler et al., 2017), which we call *Hillclimb-MLE (HC-MLE)* .

---

**Algorithm 1** Hillclimb-MLE Training Method for fitness functions

---

**Require:** pretrained MLE model $\pi_\theta$; a reward function $R$; initial set of sequences $\Sigma = \emptyset$
1: **for** n-steps **do**
2:      **for** m-sequences **do**
3:          Generate a sequence $Y_{i,1:T} = (y_1, \ldots, y_T) \sim \pi_\theta$
4:          Calculate $r_i = R(Y_{i,1:T})$
5:      **end for**
6:      $\Sigma \leftarrow \Sigma \cup \bigcup_{i=1}^{m} \{Y_i\}$
7:      Keep $k$-top sequences in $\Sigma$ with highest corresponding $r_i$
8:      Fine-tune MLE model $\pi_\theta$ to minimize cross-entropy according to Eq. 9 on dataset $\Sigma$
9: **end for**

---

### 4.1.1 REINFORCEMENT LEARNING

Recently, advances in reinforcement learning (RL) have prompted explorations into using RL methods within the drug discovery loop (Jaques et al., 2017; Olivecrona et al., 2017; Segler et al., 2017). RL is a natural environment for drug discovery, which requires online learning balanced against expensive sample evaluation and generation. Moreover, most molecule characteristics are not directly-optimizable since they are non-differentiable quantities. Instead, casting it into a RL framework

allows for the exploration of chemical space in the absence of true loss gradients. Formally, a SMILES string is a sequence $y$:

$$y = (a_1, \ldots, a_m) \qquad a \in D \tag{10}$$

Thus, each string is comprised of a set of $m$ symbols from the symbol dictionary $D$, including the padding character, establishing the SMILES environment as a large but discrete state space of size $|D|^m$ and a discrete action space of size $|D|$. From an RL perspective, each symbol in the SMILES sequence corresponds to an action $a_t$ taken at sequence step $t$. The goal of an RL agent here is to develop a policy $\pi_\theta$, parameterized by $\theta$, to calculate an action $a_t$ from the current state $s_t = (a_1, \ldots, a_{t-1})$ that maximizes the expected reward. That is, to maximize the following:

$$\mathbb{E}[R(y_{1:t})|s_0, \theta] = \sum_{y \in Y} \sum_{a_t \in y} \pi_\theta(a_t|y_{1:t-1}) \cdot Q(a_t, y_{1:t-1}) \tag{11}$$

where our policy predicts the probability of choosing an action $a_t$ in sequence $y$ from the set of all possible sequences $Y$, and the reward function $Q(a_t, y_{1:t-1})$ provides the reward for that action given the sequence so far, $y_{1:t-1}$. However, molecule generation is inherently episodic, producing rewards only at the completion of the SMILES string (in our dictionary, `'\n'`); for example, parentheses may be open which results in an invalid molecule for evaluation until the matching parentheses closes the branch.

To acquire a training gradient to use with a neural network as $\pi_\theta$, the log-derivative trick can be used to arrive at:

$$\nabla_\theta \mathbb{E}_{y_{1:t} \sim \pi(y|\theta)}[R(y)] = \mathbb{E}_{y_{1:t} \sim \pi(y|\theta)}[R(y) \log(\pi_\theta(y))] \tag{12}$$

which yields the differentiable loss used in the REINFORCE algorithm (Williams, 1992), used here as the *policy gradient* model:

$$L^{PG}(\theta) = \hat{\mathbb{E}}_t[\log \pi_\theta(a_t|s_t)r(a_t)] \tag{13}$$

$$r(a_t) = R(Y_{1:T}), \quad a_t \in Y \tag{14}$$

Notably, the final reward $R(Y_{1:t})$ is distributed equally without temporal discounting to every action in the sequence, as temporal distance in SMILES strings is less relevant than most action-spaces.

**Reinforced Generative Adversarial Networks**  The recently-introduced GANs (Goodfellow et al., 2014) have experienced a surge of popularity, extending in new ways their original formulation:

$$\min_G \max_D V(D, G) = \mathbb{E}_{x \sim p_{\text{data}}(x)}[\log D(x)] + \mathbb{E}_{z \sim p_z(z)}[\log(1 - D(G(z)))] \tag{15}$$

which describes the minimax zero-sum game that combines a generator $G$, converting samples $z$ from a standard distribution $p_z$ into data-like samples, and a discriminator $D$ which attempts to determine whether samples $x$ are from the true distribution $p_{data}$ and which are generated samples $G(z \sim p_z)$. GANs have been applied as a novel architecture type to a variety of domains, particularly in image synthesis where they have been found to generate perceptually realistic images (Radford et al., 2015; Zhu et al., 2017).

The GAN formulation was extended in Yu et al. (2017) to allow sequences to be generated with a GAN-like architecture, using RL and a policy gradient technique to train. This forms the basis of the method published in Guimaraes et al. (2017) to use a RL framework with GANs to optimize arbitrary fitness functions. Training is comprised of three phases: a maximum-likelihood pretraining phase during which the RNN generator learns to generate molecules from a chemical structure dataset, while a discriminator is pretrained on produced *versus* within-dataset sequences; a RL generator phase in which the generator learns to maximize the reward function balanced with the discriminator; and, finally, a discrimination phase which subsequently guides generation. The latter two phases are alternated throughout the optimization phase.

The reward used to reinforce the network uses a parameter $\lambda$ which balances the likelihood that the discriminator $D_\phi$ classifies the generated sequence as true data against the arbitrary fitness objective $O$:

$$R(Y_{1:T}) = \lambda D_\phi(Y_{1:T}) + (1 - \lambda)O(Y_{1:T}) \tag{16}$$

However, while calculating the reward during sequence generation, the final reward is not yet available until the end of the sequence. In order to maximize the number of training steps (rather than one step per sequence) and to guide the generation process, the objective-reinforced GAN (Guimaraes et al., 2017) uses Monte-Carlo rollouts from the current state to estimate the reward of the available actions. This strategy, however, can have high variance and significantly undersamples the large branching in molecule generation (with 42 symbols per step and an average sequence length of 44 steps).

**Advantage Actor-Critic Networks**    A wealth of new techniques for reinforcement learning have recently appeared which have yet to be adapted to molecule generation. One technique to minimize the variance that hinders policy gradient training and to accelerate convergence is to subtract an estimate of the reward from the true reward. Importantly, this does not introduce estimator bias but can diminish the variance. This can be seen as an actor-critic model that separates the policy training from the value estimation in which the actor is the policy $\pi_\theta$ and the baseline value estimate $V_{s_t}$ is the critic that gives the approximate value of that state (Degris et al., 2012; Sutton & Barto, 1998; Mnih et al., 2016). This *advantage actor-critic* (A2C) training is the synchronous version of the asynchronous advantage actor critic (A3C) model (Mnih et al., 2016) and can use generalized advantage estimation (Schulman et al., 2015). In this work, the following differentiable loss function is used to simultaneously reduce the variance of the reward and to minimize the difference between the expected reward $V_\phi$ and the true reward:

$$L^{A2C}(\theta, \phi) = \hat{\mathbb{E}}_t[\log \pi_\theta(a_t|s_t)(R - V_\phi(s_t)) + (R(s_t) - V_\phi(s_t))^2] \tag{17}$$

The value estimation function $V_\phi$ is a neural network itself here, parameterized by $\phi$; this work uses a fully-connected network from the hidden state of the LSTM to a single node predicting reward.

**Proximal Policy Optimization**    Finally, great success has been found recently in the family of proximal policy optimization algorithms (Schulman et al., 2017). These algorithms exhibit very low sample complexity and are of particular interest as reliable data is always at a premium in chemistry, and both generation and evaluation time of a molecule can be long. Using the advantage $\hat{A}_t = R(s_t) - V_\phi(s_t)$ from above, the following clipped PPO loss is used:

$$L^{CLIP}(\theta, \phi) = \hat{\mathbb{E}}_t \left[ \min \left( \frac{\pi_\theta(a_t|s_t)}{\pi_{\theta^{old}}(a_t|s_t)} \hat{A}_t, \text{clip} \left( \frac{\pi_\theta(a_t|s_t)}{\pi_{\theta^{old}}(a_t|s_t)}, 1 + \epsilon, 1 - \epsilon \right) \hat{A}_t \right) + \hat{A}_t^2 \right] \tag{18}$$

The motivation for this pessimistic loss given by the min() function, succinctly stated in Schulman et al. (2017), ignores changes in the probability ratio when it would increase the objective and focuses learning in which the objective would worsen. Furthermore, the clipped boundary region provides a tight bound, outside of which the loss has function has no incentive to further change, encouraging smaller changes. The hyperparameter $\epsilon$ controls the width of this region and can be found in Table 4.

**Regularization for Molecular Generation**    There are additional costs that can be combined with the above techniques. As the goal of *de novo* design is not to generate a single molecule, but rather a family of candidate molecules to present to an expert domain scientist; a diverse set of suggested molecules is always beneficial. To encourage robust learning, a regularization factor on the Kullback-Leibler (KL) divergence between the current policy and the original policy can be added to encourage the model to not stray far from a good model of chemical distributions:

$$C^{MLE} = \lambda_{MLE}\hat{\mathbb{E}}_t \left[ \sum_{a_i \in D} \pi_\theta(a_i|s_t) \log \frac{\pi_\theta(a_i|s_t)}{\pi_\theta^{MLE}(a_i|s_t)} \right] \tag{19}$$

using a weight parameter $\lambda$ and evaluating the static, pretrained policy $\pi_\theta^{MLE}(a_i|s_t)$. Similarly, for RL scenarios, a way to explicitly encourage exploration is to place a cost on the distribution entropy of the model:

$$C^{ENT} = \lambda_{ent}\hat{\mathbb{E}}_t \left[ -\sum_{a_i \in D} \pi_\theta(a_i|s_t) \log \pi_\theta(a_i|s_t) \right] \tag{20}$$

The values used in this work can be found in the supplementary material. Additionally, we evaluate the policy gradient regularization scheme proposed by Olivecrona et al. (2017), denoted "Aug. MLE" in this paper.

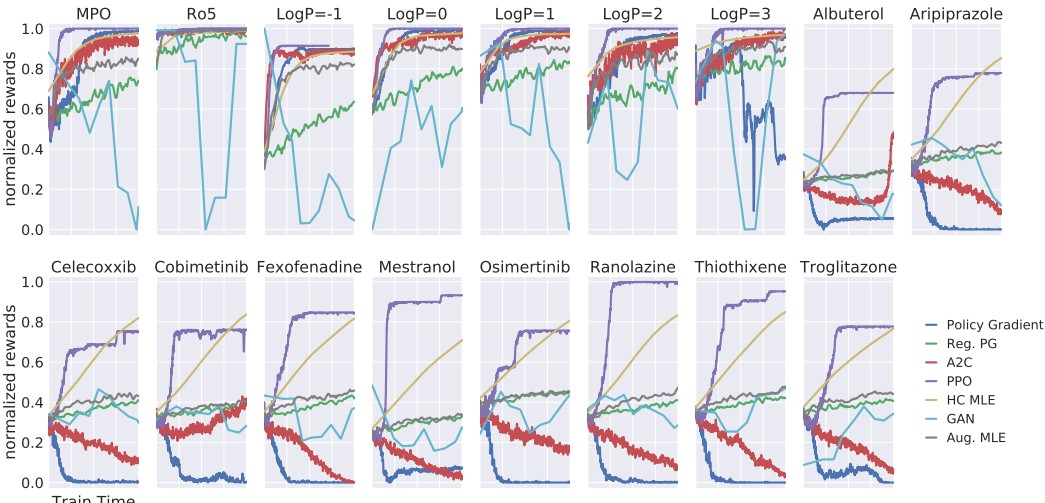

Figure 3: Sample RL trajectories for the tasks (independently-scaled).

## 5 RESULTS

These algorithms were built using PyTorch (Paszke et al., 2017), with RDKit (Landrum et al.) providing the chemoinformatics functionality. These benchmarks were built as new environments in the OpenAI Gym (Brockman et al., 2016), and will be released as open-source to the community. The code was run using an NVIDIA Tesla V100 GPU, with optimization times for 1000 action-steps shown in the "Runtime" category of Table 1. The final RNN architecture that underlies the Policy Gradient, Regularized PG, GAN, A2C, PPO, HC MLE, and Augmented MLE models consists of three layers of 512 LSTM neurons, with a 512-dimensional embedding layer and a fully-connected output layer of 42 neurons. It takes approximately eight hours to train for 70 epochs on the filtered ChEMBL dataset, with 50,000 randomly-sampled SMILES strings for development, 200,000 for test, and the remaining 1190203 used for training. For the GAN architecture, the discriminator is a convolutional network with two alternating layers of ten kernels of 3x3 convolutions followed by 2x2 max-pooling, with global averaging over the final two (real and fake) outputs. Experiments were run for approximately 1.6M action-steps, corresponding roughly to 1000 SMILES in batches of 32 with an average sequence length of 44.

Encouragingly, the baseline MLE model after training produces 94.7% valid SMILES out of 10k generated molecules without further fine-tuning, establishing an extremely competitive benchmark. For similar evaluation, a GVAE was trained on the same dataset and achieved 37.1% valid molecules. Moreover, 99.87% of these MLE-sampled molecules were unique, establishing very high diversity.

The RL model results, which are the remaining 17 benchmark tasks including multi-objective optimization, Ro5 optimization, ClogP targeting, and drug-fingerprint targeting, can be found in Table 1, with example RL curves in Fig. 3. The benchmarks referred to by a drug name are the Tanimoto-approach benchmarks, in which a drug is held out from the training set and the model must design a similar drug given only the FCFC4 semantic hash of the drug. The baseline model samples 10k random molecules from the training set, keeping only the best-performing ones.

**Hillclimb-MLE**    Perhaps surprisingly, the alternating sampling and HC-MLE training algorithm was the most successful model. The model effectively avoided nonoptimal local minima and steadily increased reward throughout training. Given a large computation budget, this model appears the most successful for widely sampling while still optimizing towards a target.

**PPO**    The PPO with the clipped objective was the most successful standard RL algorithm for this task. It often converges an order-of-magnitude faster than other algorithms, but occasionally chooses a less optimal final configuration or occasional catastrophic loss. For future work where molecule fitness function testing time is significant - if, for example, an assay is required - PPO is a valuable

Table 1: Model performance, given by mean fitness in the final timestep over three random initializations, while single-best SMILES result from the plotted runs is given in parentheses.

| | | Baseline | Reg. PG | A2C | PPO | HC-MLE |
|---|---|---|---|---|---|---|
| Property | LogP=-1 | 1.00 | 0.66 (1.00) | 0.98 (1.00) | 1.00 (1.00) | 0.97 (1.00) |
| | LogP=0 | 1.00 | 0.78 (1.00) | 0.98 (1.00) | 1.00 (1.00) | 0.98 (1.00) |
| | LogP=1 | 1.00 | 0.83 (1.00) | 0.98 (1.00) | 1.00 (1.00) | 0.97 (1.00) |
| | LogP=2 | 1.00 | 0.86 (1.00) | 0.97 (1.00) | 1.00 (1.00) | 0.97 (1.00) |
| | LogP=3 | 1.00 | 0.86 (1.00) | 0.97 (1.00) | 0.91 (1.00) | 0.97 (1.00) |
| Mult. Obj. | MPO | 1.00 | 0.82 (1.00) | 0.95 (1.00) | 1.00 (1.00) | 0.98 (1.00) |
| | Ro5 | 1.00 | 0.77 (1.00) | 0.96 (1.00) | 1.00 (1.00) | 0.59 (1.00) |
| Tanimoto | Albuterol | 0.02 | -0.55 (0.41) | 0.14 (-0.08) | 0.04 (-0.10) | 0.32 (0.83) |
| | Aripiprazole | -0.15 | -0.34 (0.63) | 0.38 (-0.12) | 0.40 (0.29) | 0.51 (1.00) |
| | Celecoxxib | -0.22 | -0.35 (0.69) | 0.20 (-0.06) | 0.25 (0.14) | 0.43 (1.00) |
| | Cobimetinib | -0.18 | -0.47 (0.17) | -0.01 (-0.01) | 0.11 (0.06) | 0.32 (0.57) |
| | Fexofenadine | -0.26 | -0.33 (0.50) | -0.24 (-0.13) | 0.18 (0.19) | 0.47 (0.82) |
| | Mestranol | -0.17 | -0.46 (0.62) | 0.14 (-0.22) | 0.06 (0.30) | 0.34 (0.85) |
| | Osimertinib | -0.44 | -0.43 (0.15) | -0.36 (-0.26) | -0.11 (0.11) | 0.13 (0.48) |
| | Ranolazine | -0.20 | -0.32 (0.49) | 0.32 (-0.19) | 0.14 (0.47) | 0.50 (1.00) |
| | Thiothixene | -0.26 | -0.35 (0.28) | -0.09 (-0.19) | 0.07 (0.29) | 0.33 (0.57) |
| | Troglitazone | -0.28 | -0.39 (0.27) | -0.19 (-0.27) | 0.06 (0.18) | 0.24 (0.56) |
| Summary | Mean | 0.30 | 0.09 (0.66) | 0.42 (0.32) | 0.48 (0.53) | 0.59 (0.81) |
| | Runtime | 0.025s | 0.68s | 2.5s | 8.54s | 0.31s |

candidate as an optimization strategy. A wide range of learning rates and entropy exploration costs were sampled (not shown) with PPO consistently and reliably training to achieve good performance.

**GAN** The GAN is a nascent and intriguing architecture, but was found to not be optimal for this task yet. The complexity of balancing the discriminator against the generator, interleaving epochs, and choosing a correct architecture made it difficult to use, though a significant body of literature is working to address more stable training (Arjovsky et al., 2017; Salimans et al., 2016). Though MC rollouts are costly in action-steps (leading to the slightly unfair showing in Figure 3), the other problem is more fundamental: as Fig. 4a shows in a Principal Component Analysis (PCA) plot of the MQN descriptors (Nguyen et al., 2009) of the generated molecules, stronger discriminator weight $\phi_D$ forces the generated molecules to more closely match the primary modes of the training data. Effectively reproducing the training data, however, contrasts with the purpose of the generator for optimizing towards a given molecule. While this operates effectively as a regularizer, it is far more stable and explicit to regularize against the KL divergence as is done for the Regularized Policy Gradient model than to balance the hyperparameters of a GAN. Perhaps future work can investigate other uses of the GAN to, *e.g.*, coerce the architecture away from reproducing identical sequences rather than memorizing the training data.

**Benefits of extended sampling** Examining the difference in the best single reward on the plotted runs (parentheses) in Table 1 and the final timestep mean (no parentheses), algorithms clearly benefit merely from sampling from a well-trained initial model for an extended period of time. The regularized policy gradient model, in particular, demonstrates high peak performance yet consistently does not learn effectively, suggesting that straightforward sampling for a large, diverse model may be an acceptable strategy in the absence of convergence-time costs.

**Temperature Sampling** A temperature factor is often included in available implementations to allow the generation of more diverse molecules. It has a straightforward formulation for an output distribution $P$ yielding $P_t = \frac{1}{Z}e^{P/t}$, with normalization constant $Z$, temperature $t$, and temperature-altered probability distribution $P_t$. However, as shown in the t-distributed Stochastic Neighbor Embedding (t-SNE) (Maaten & Hinton, 2008) plot in Fig. 4b, higher temperatures do not necessarily translate to a better coverage of chemical space. As a large proportion of possible SMILES strings are invalid, increasing the temperature is likely to invalidate the SMILES string. The molecules generated

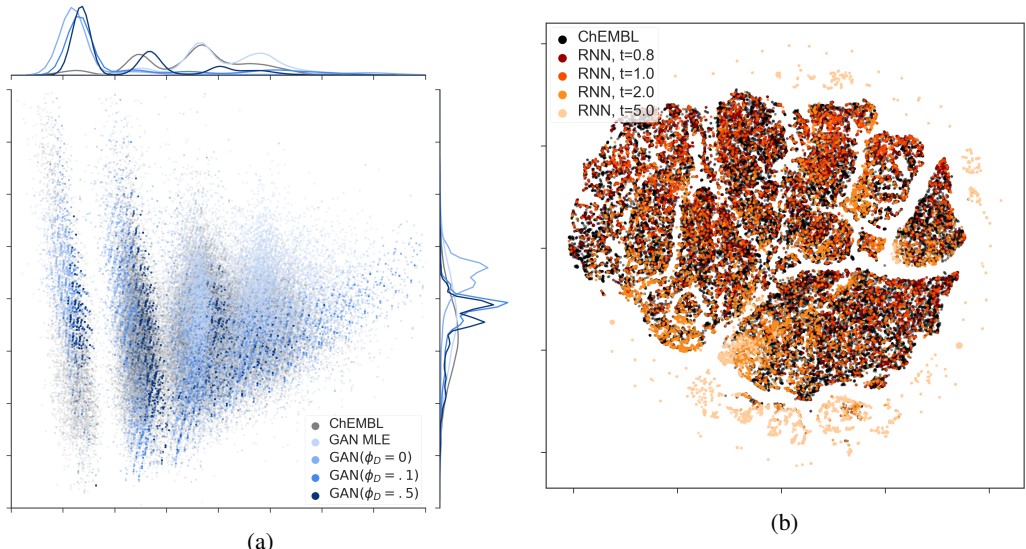

(a)                                    (b)

Figure 4: In **(a)**, PCA plot of the GAN generator under various influences of the discriminator. In **(b)**, sampling from a model at higher temperatures counter-intuitively does not increase coverage.

in the highest-temperature network favor short, brief SMILES strings that vastly undersample the chemical space. Indeed, this suggests the best strategy to achieve high diversity and a larger sampling of valid space is to not use temperature sampling at all.

## 6    CONCLUSION

In this work, we proposed a large, standardized dataset and a set of 19 benchmarks to evaluate models for molecule generation and design. Several RL strategies were investigated on these benchmarks. Here, the results suggest that the Hillclimb-MLE model is a surprisingly robust technique with large datasets and deep models, outperforming PPO given sufficient compute times and sample evaluations. In the space of constrained compute and sample evaluations, PPO was shown to be an effective learning algorithm, converging an order of magnitude before other reinforcement-learning algorithms for molecule generation.

Nevertheless, there is still tremendous need for more efficient and effective models for molecular design, which could have a profound impact on molecular design — including drug, materials and agrochemicals discovery — and thus immediately on human well-being. With the present, easily usable benchmark, we hope to inspire the machine learning community to pick up this challenge.

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

## A    SUPPLEMENTARY MATERIAL

This supplementary material section offers additional information and to aid in reproducibility for the reader. For visibility regarding the extent to which the MLE pretrained-model matches the trained space of chemistry, Figure 5 demonstrates its ability to effectively map and generate the space of ChEMBL molecules. In Table 2, the SMILES dictionary used in this work can be found. The drugs used in the Tanimoto fingerprint approach task can be found in Table 3, with the hyperparameters used in this work in Table 4.

### DATASET PREPROCESSING

Preprocessing the Chembl-23 dataset is comprised of a few steps. First, all molecules containing the compounds are neutralized and salts are stripped, keeping only the largest connected component. Then, the canonical SMILES representation of the molecules are generated using RDKit (Landrum et al.), and all SMILES sequences longer than 100 characters are filtered out. Most molecules with more than 100 SMILES symbols are either peptides, too heavy or too complex to be considered in *de novo*-design, and 96.7% of ChEMBL is less than 100 symbols in length. Any SMILES containing forbidden symbols (Table 2) are filtered out. Further, the molecules of the Tanimoto benchmark are removed from the set. Finally, all molecules similar to these benchmark molecules in terms a Tanimoto coefficient greater than 0.5 on ECFC4 fingerprints are removed. The remaining molecules are randomly split into train, test and development set.

Before one-hot encoding, the four multi-character included atoms (Br, Cl, Si, Se) were replaced with single-character substitutes. Then, the included symbols (including start character '$Q$' and end character '\n' are converted to integer token equivalents.

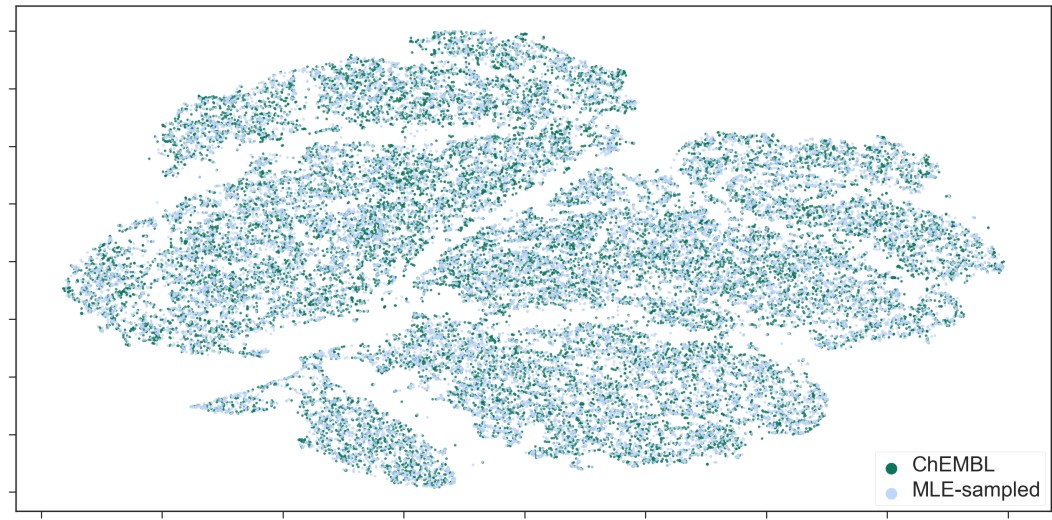

Figure 5: t-SNE visualization (Maaten & Hinton, 2008) of MLE sampling of generated space. The MLE model effectively covers the space of ChEMBL and even reproduces the subspaces around the ChEMBL molecules.

Table 2: Dictionary of SMILES

| Forbidden Symbols | 'Li','Be','Ne','Na','Mg','Al','Ar','K','Ca','Sc', 'Ti','V','Cr','Mn','Fe','Co','Ni','Cu','Zn','Ga', 'Ge','As','Kr','Rb','Sr','Y','Zr','Nb','Mo','Tc', 'Ru','Rh','Pd','Ag','Cd','In','Sn','Sb','Xe','Cs', 'Ba','Hf','Ta','W','Re','Os','Ir','Pt','Au','Hg', 'Tl','Pb','Bi','Po','At','Rn','Fr','Ra','Rf','Db', 'Sg','Bh','Hs','Mt','Ds','Rg','Cn','Fl','Lv','La', 'Ce','Pr','Nd','Pm','Sm','Eu','Gd','Tb','Dy','Ho', 'Er','Tm','Yb','Lu','Ac','Th','Pa','U','Np','Pu', 'Am','Cm','Bk','Cf','Es','Fm','Md','No','Lr','as', 'te','Te','se' |
|---|---|
| Multi-character Symbols | 'Br':Y, 'Cl':X, 'Si':A, 'Se':Z |
| Included Symbols | 'Q','\n',' ','#','%','(', ')','+','-','.','0','1', '2','3','4','5','6','7','8','9','=','A','B','C', 'F','H','I','N','O','P','S','X','Y','Z','[',']', 'b','c','n','o','p','s' |

Table 3: Drugs used in the Tanimoto benchmark task

| Drug | SMILES |
|---|---|
| Albuterol | CC(C)(C)NCC(O)c1ccc(O)c(CO)c1 |
| Aripiprazole | Clc1cccc(N2CCN(CCCCOc3ccc4CCC(=O)Nc4c3)CC2)c1Cl |
| Celecoxxib | Cc1ccc(cc1)-c1cc(nn1-c1ccc(cc1)S(N)(=O)=O)C(F)(F)F |
| Cobimetinib | OC1(CN(C1)C(=O)c1ccc(F)c(F)c1Nc1ccc(I)cc1F)C1CCCCN1 |
| Fexofenadine | CC(C)(C(O)=O)c1ccc(cc1)C(O)CCCN1CCC(CC1)C(O)(c1ccccc1)c1ccccc1 |
| Mestranol | COc1ccc2C3CCC4(C)C(CCC4(O)C#C)C3CCc2c1 |
| Osimertinib | COc1cc(N(C)CCN(C)C)c(NC(=O)C=C)cc1Nc2nccc(n2)c3cn(C)c4ccccc34 |
| Ranolazine | COc1ccccc1OCC(O)CN2CCN(CC(=O)Nc3c(C)cccc3C)CC2 |
| Thiothixene | CN(C)S(=O)(=O)c1ccc2Sc3ccccc3C(=CCCN4CCN(C)CC4)c2c1 |
| Troglitazone | Cc1c(C)c2OC(C)(COc3ccc(CC4SC(=O)NC4=O)cc3)CCc2c(C)c1O |

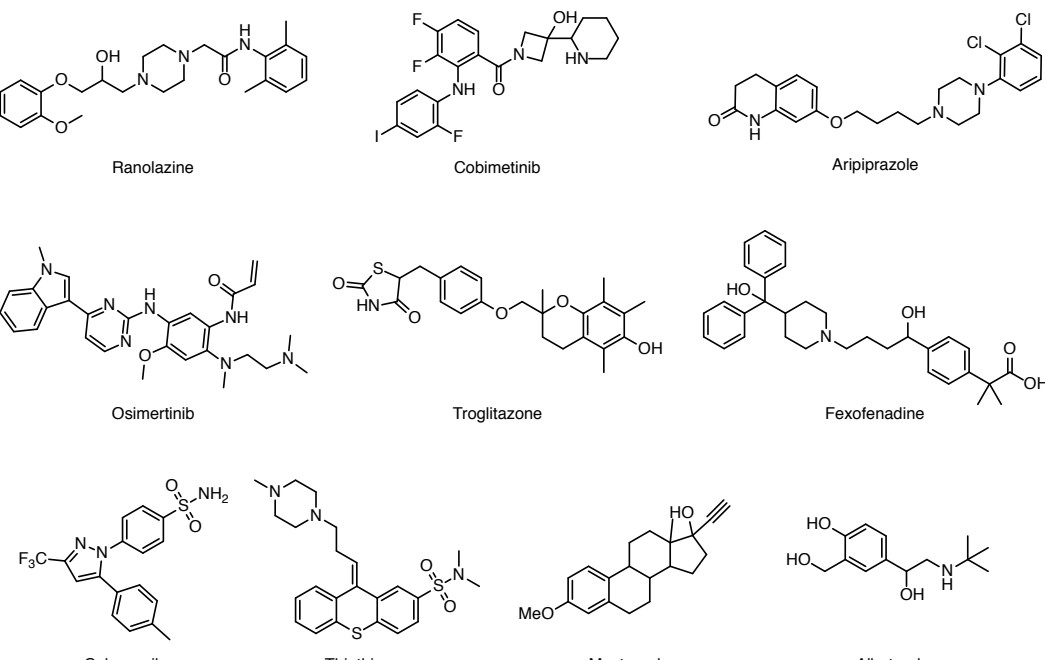

Figure 6: Target molecules for the Tanimoto benchmark.

Table 4: Hyperparameters

| Model | Param | Explanation |
|---|---|---|
| Baseline | `batchsize = 10` | Number of SMILES per batch |
| | `epochs = 1000` | Number of batches |
| Reg. PG | `max_len=100` | Max length of a SMILES |
| | `batchsize=128` | Number of parallel batches |
| | `num_epochs=128` | Number of batches of RL optimization |
| | `c_reg=10` | Weight of the MLE regularizer |
| | `lr=1e-4` | Learning rate of Adam optimizer |
| A2C | `num_actions=1e6` | Total number of actions of RL optimization |
| | `max_grad_norm=0.5` | Maximum norm of gradient before clipping |
| | `gamma=1.0` | Time-discount factor |
| | `lr=4e-4` | Learning rate of Adam optimizer |
| | `eps=1e-8` | Epsilon parameter for Adam optimizer |
| | `num_steps=40` | Number of steps of forward steps |
| | `batch_size=32` | Number of parallel environments |
| | `entropy_coef=0.013` | Multiplicative cost of the entropy |
| | `gae=1` | Use generalized advantage estimation |
| | `tau=0.95` | Generalized advantage estimation tau |
| | `val_loss_wt=0.5` | Weight of the value loss |
| GAN | `pretrain_discrim=5` | Number of epochs to pretrain the discriminator |
| | `num_rounds=4` | Number of rounds of RL-and-discriminator alternating training |
| | `rl_epochs=5` | Number of RL epochs per round |
| | `discr_epochs=1` | Number of discriminator epochs per round |
| | `phi_D=0.05` | Weight of the discriminator in reward |
| | `batch_size=8` | Batch size of training |
| | `max_len=100` | Max rollout length |
| | `gamma=0.98` | Time-discount factor for rollouts |
| | `num_rollouts=3` | Number of rollouts per action-step |
| | `lr=1e-3` | Learning rate of Adam optimizer for generator |
| | `lr=1e-3` | Learning rate of Adam optimizer for discriminator |
| PPO | `ppo_batch_size=64` | Batch size of the PPO updates |
| | `ppo_epoch=4` | Number of PPO epochs |
| | `epsilon=0.2` | Width of the PPO clip region |
| | `num_actions=1e6` | Total number of actions of RL optimization |
| | `max_grad_norm=0.5` | Maximum norm of gradient before clipping |
| | `gamma=1.0` | Time-discount factor |
| | `lr=4e-4` | Learning rate of Adam optimizer |
| | `eps=1e-8` | Epsilon parameter for Adam optimizer |
| | `num_steps=40` | Number of steps of forward steps |
| | `batch_size=32` | Number of parallel environments |
| | `entropy_coef=0.013` | Multiplicative cost of the entropy |
| | `gae=0` | Do not use generalized advantage estimation |
| | `val_loss_wt=0.5` | Weight of the value loss |

