# OpenReview forum: "Exploring Deep Recurrent Models with Reinforcement Learning for Molecule Design"
_ICLR.cc/2018/Conference — Invite to Workshop Track_

### Official Review · AnonReviewer2 · 2017-11-09

**Rating:** 4
**Confidence:** 2

**Review:**

The paper proposes a set of benchmarks for molecular design, and compares different deep models against them. The main contributions of the paper are 19 molecular design benchmarks (with chembl-23 dataset), including two molecular design evaluation criterias and comparison of some deep models using these benchmarks. The paper does not seem to include any method development.

The paper suffers from a lack of focus. Several existing models are discussed to some length, while the benchmarks are introduced quite shortly. The dataset is not very clearly defined: it seems that there are 1.2 million training instance, does this apply for all benchmarks? The paper's title also does not seem to fit: this feels like a survey paper, which is not reflected in the title. Biologically lots of important atoms are excluded from the dataset, for instance natrium, calcium and kalium. I don't see any reason to exlude these. What does "biological activities on 11538 targets" mean?

The paper discussed molecular generation and reinforcement learning, but it is somewhat unclear how it relates to the proposed dataset since a standard training/test setting is used. Are the test molecules somehow generated in a directed or undirected fashion? Shouldn't there also be experiments on comparing ways to generate suitable molecules, and how well they match the proposed criterion? There should be benchmarks for predicting molecular properties (standard regression), and for generating molecules with certain properties. Currently it's unclear which type of problems are solved here.

Table 1 lists 5 models, while fig 3 contains 7, why the discrepancy? In table 1 the plotted runs seem to differ a lot from average results (e.g. -0.43 to 0.15, or 0.32 to 0.83). Variances should be added, and preferably more than 3 initialisations used.

Overall this is an interesting paper, but does not have any methodological contribution, and there is also few insightful results about the compared methods, nor is there meaningful analysis of the problem domain of molecules either.

---

> ### Author Response · Authors · 2018-01-05
> **We have adapted the manuscript to make the contributions and scope clearer**
>
>
> We thank the reviewer for their critical feedback. We have adapted the manuscript to make the contributions and scope clearer.
>
> Algorithms that allow the generation of small molecules (small graphs) that satisfy given desirable properties are rapidly evolving for medicine, materials and agriculture.  This is currently an area of intense investigation, theoretically as well as practically, as highlighted by several ICLR submissions on molecule/graph generation this year, which all employ different and inconsistent benchmarks and training sets. Currently, it is not possible to compare these results.
>
> With this paper, we unify the problem space of small molecule generation and thoroughly investigate various approaches (including algorithms which have yet to be explored in this domain, e.g. PPO), and include evaluations of our new benchmarks on pre-existing work.
>
> Our results surpass state-of-the-art results previously reported on a few of the sub-domains, establishing new baselines, and come to the perhaps surprising result that the relatively simple hill-climbing MLE method achieves results on par with the some of the most advanced recently-developed RL algorithms such as PPO.
>
>
> Specific Comments:
>
> * Regarding lack of focus: we have reorganized the paper to improve the clarity of the work.
>
> * Regarding purpose: We hope the above responses address your concern of this being a survey work. We believe this work introduces new benchmarks, evaluates pre-existing algorithms, demonstrates novel pairings of algorithm and domain, and establishes a new state-of-the art as well as a few surprising additional insights (hill-climbing MLE supremacy, temperature ineffectiveness).
>
> * Regarding preprocessing steps: the steps taken in this work are standard and in line with the field of computational chemistry [1,2]. This includes the removal of Sodium, Calcium and Potassium, and other counterions.
>
> * Regarding clarity of RL vs. train/test set: for algorithms that rely on pretraining to help navigate the extremely large space of small molecule generation (~10^60), it is important that the algorithms have not been exposed to a correct solution in the training set, hence the train-test split.  As an additional benefit, a train/test split permits the benchmarks to be used with rule-based GOFAI systems, supervised algorithms as well as RL.
>
> * Regarding molecular property prediction: indeed, this is an important sub-field of computational chemistry and is explored under the family of QSAR models [3, 4].  However, that is out-of-scope of this paper, as it attempts to address a different concern.
>
> * Regarding data: the table is a bit overwhelming already, so we chose not to exhaustively show all results for all models and instead focus on representative key models.  Due to time and computational constraints, we had not run more than three initializations, but can do so for the revision.
>
>
> We hope this addresses your concerns.
>
> [1] Glaab, Enrico. "Building a virtual ligand screening pipeline using free software: a survey." Briefings in bioinformatics 17.2 (2015): 352-366.
> [2] Lionta, Evanthia, et al. "Structure-based virtual screening for drug discovery: principles, applications and recent advances." Current topics in medicinal chemistry 14.16 (2014): 1923-1938.
> [3] Tropsha, Alexander. "Best practices for QSAR model development, validation, and exploitation." Molecular informatics 29.6‐7 (2010): 476-488.
> [4] Tropsha, Alexander, and Alexander Golbraikh. "Predictive QSAR modeling workflow, model applicability domains, and virtual screening." Current pharmaceutical design 13.34 (2007): 3494-3504.

---

### Official Review · AnonReviewer1 · 2017-11-27
**This is a solid paper about model evaluation in the chemical domain.**

**Rating:** 7
**Confidence:** 4

**Review:**

Summary:
This work is about model evaluation for molecule generation and design. 19 benchmarks are proposed, small data sets are expanded to a large, standardized data set and it is explored how to apply new RL techniques effectively for molecular design.

on the positive side:
The paper is well written, quality and clarity of the work are good. The work provides a good overview about how to apply new reinforcement learning techniques for sequence generation. It is investigated how several RL strategies perform on a large, standardized data set. Different RL models like Hillclimb-MLE, PPO, GAN, A2C are investigated and discussed.  An implementation of 19 suggested benchmarks of relevance for de novo design will be provided as open source as an OpenAI Gym.


on the negative side:
There is no new novel contribution on the methods side.



minor comments:

Section 2.1.
see Fig.2 —> see Fig.1
page 4just before equation 8: the the

---

> ### Author Response · Authors · 2018-01-05
> **Reply**
>
>
> We are grateful for your comments.  We hope your concern about novelty is addressed with our main comment; indeed, the pairing here is in the algorithm to this particular application area.
> We further hope that a foundational framework proposed will allow the emergence of future, novel algorithms.  Your minor comments have been addressed in the manuscript.

---

### Official Review · AnonReviewer3 · 2017-11-27
**empirical evaluation of recurrent models and RL for molecule design**

**Rating:** 6
**Confidence:** 3

**Review:**

Summary: This paper studies a series of reinforcement learning (RL) techniques in combination with recurrent neural networks (RNNs) to model and synthesise molecules. The experiments seem extensive, using many recently proposed RL methods, and show that most sophisticated RL methods are less effective than the simple hill-climbing technique, with PPO is perhaps the only exception.

Originality and significance:

The conclusion from the experiments could be valuable to the broader sequence generation/synthesis field, showing that many current RL techniques can fail dramatically.

The paper does not provide any theoretical contribution but nevertheless is a good application paper combining and comparing different techniques.

Clarity: The paper is generally well-written. However, I'm not an expert in molecule design, so might not have caught any trivial errors in the experimental set-up.

---

> ### Author Response · Authors · 2018-01-05
> **Reply**
>
> We thank the referee for their comments and perspective on our work.
>
> We hope this reviewer’s comments have been addressed in our overall reply and in the responses to the other reviewers.

---

### Author Response · Authors · 2018-01-05
**Global Reply**


We thank the reviewers for their effort and advice towards improving our submission.  We are pleased that the reviewers have identified our important contributions of dataset curation and preprocessing steps, proposed benchmarks, and baseline results using recently-developed algorithms.  While we introduce no new reinforcement learning algorithms in this work, our primary aim was to substantially lower the barrier towards automated molecular design to allow computer scientists with no prior background in chemistry to develop novel algorithms to improve molecular design.  Indeed, here the novelty lies in the pairing of task and algorithm, and this work is foundational to clearly lay out steps and provide code to apply reinforcement learning algorithms to molecule design.

Finally, we are able to demonstrate results in this manuscript that establish a new state-of-the-art in single and multi objective physicochemical property optimization and chemical space exploration tasks.  Subsequent work can then build on this set of standardized molecular design benchmarks to introduce new methods. The benchmark framework is general enough to be used with any possible small molecule generation method, whether rule-based or learned, and is not limited to sequence-based generation relying on SMILES. We have therefore amended the manuscript to reflect the importance of our introduced benchmarks.

---

### Public Comment · (anonymous) · 2018-01-17
**Relevant paper not cited**

There is a recently published paper on molecular design with deep reinforcement learning which is not addressed in your work:
Popova, Mariya, Olexandr Isayev, and Alexander Tropsha. "Deep reinforcement learning for de-novo drug design." arXiv preprint arXiv:1711.10907 (2017).
This paper discusses alternative to GANs method of novel compounds generation with recurrent neural network and reinforce algorithm, which is relevant to your work.

---

> ### Author Response · Authors · 2018-01-18
> **"Relevant" paper was published one month after ours.**
>
> The first version our paper was uploaded to openreview on 27th Oct 2017.
>
> The paper by Popova et al. was put on arXiv on 29th Nov 2017, that is about a month later than our paper.

---

### Public Comment · ~Mostapha_Benhenda1 · 2018-02-24
**Narrow approach (only RL) that ignores previous literature.**

1. No discussion of previous measures of diversity in the literature, like Guimaraes et al (30 May 2017), or Benhenda (28 August 2017). There are even more diversity metrics in the recent Benhenda et al.  (8 February 2018) here: https://www.authorea.com/users/226673/articles/285209-diversitynet-a-collaborative-benchmark-for-generative-ai-models-in-chemistry

Your definition of diversity is extremely rudimentary, why is it the best one?

2. Your paper proposes 19 new tasks, but completely ignored tasks from the previous literature, like in Olivecrona et al. (25 April 2017), Dieb et al. (20 July 2017), Sanchez-Lengeling et al. (17 August 2017). If you are a staunch advocate of reproductibility, why these omissions?

On the other hand, after your paper, the literature totally ignored your platform, like Popova et al.  (29 November 2017), Ertl et al. (20 December 2017) and many others. See a list here: https://medium.com/the-ai-lab/diversitynet-a-collaborative-benchmark-for-generative-ai-models-in-chemistry-f1b9cc669cba

After 4 months passed, how do you explain the persisting lack of adoption of your platform by the community ?
In my case, I just didn’t hear about it at that time. Lack of marketing. Thanks Nathan Brown for tweeting.

3. However, your approach is too restrictive, you remain within reinforcement learning models. There are important works with autoencoders, like Gomez-Bombarelli et al. (2016), Kadurin et al. (July 2017) and many others, and they deserve to be benchmarked too. Is the OpenAI gym platform restricted to Reinforcement Learning ?

The ongoing DiversityNet molecule benchmark (here) is using Texygen platform (appeared on the  6th February 2018), which is more inclusive for non-RL algorithms. See: https://github.com/geek-ai/Texygen/issues/5
However, Texygen is not generalized yet to the very few non-SMILES generative models in the literature. That’s something to do.

4. Does multi objective optimization always succeed by taking 'any arbitrary balanced weighting' of objectives ?  Is it always so easy? A general discussion, with different objectives, would have been welcome. Otherwise, the uninformed reader could imagine that it’s a piece of cake to do Multi-objective reinforcement learning.

5. That said, your paper makes an interesting curation of data, models, and tasks. It can be useful for a proper benchmark.

In conclusion, I don’t think that the ML community should tackle this challenge the way you presented it in this paper 4 months ago.

---

### Decision · Program_Chairs · 2018-01-29
**ICLR 2018 Conference Acceptance Decision**

**Decision:**

Invite to Workshop Track

**Comment:**

The paper creates a dataset for exploration of RL for molecular design and I think this makes it a strong contribution to the community at the intersection of the two. For a methods focussed conference such as ICLR however, it may not be the best fit. Hence I would recommend submitting to a workshop track or targeting a more focussed venue such as a bioinformatics conference.